# Fluid shear stress-dependent modulation of the basal endothelial glycocalyx

**Zoe Vittum**[1], **Solomon A. Mensah**[1,2]*

**1** Biomedical Engineering Department, Worcester Polytechnic Institute, Worcester, Massachusetts, United States of America, **2** Mechanical Engineering Department, Worcester Polytechnic Institute, Worcester, Massachusetts, United States of America

* smensah@wpi.edu

## Abstract

The vascular endothelial glycocalyx is a major regulator of endothelium function, serving as a vital mechanotransducer and barrier. Shear stress due to blood flow exerted on the apical endothelial cell (EC) and glycocalyx have been the primary focus of the field due to their direct interaction. Recently, it has been demonstrated that the basal glycocalyx, exposed to the basement membrane, is sensitive to apical fluid shear stress and necessary for promoting mechanotransduction pathways key in the EC response to blood flow. Here, we demonstrate the fluid shear stress-dependent modulation of the basal endothelial glycocalyx, showing a notably different expression pattern and overall increased expression compared to the apical glycocalyx. Our findings, coupled with prior evidence linking the basal GCX to cytoskeletal dynamics, underscore its potential role in vascular barrier integrity and mechanotransduction.

## Introduction

The endothelial glycocalyx (GCX) is one of the major regulators of endothelium function. The apical GCX or the GCX that lines the luminal surface of ECs, acts as a dynamic interface between extracellular stimuli and the EC, modulating how the EC transduces and responds to biophysical and biochemical signals [1–5]. Interactions with physiological stimuli ideally result in a robust apical GCX capable of extracellular force transduction and regulation of vascular permeability [3,4,6–8]. However, the apical GCX can become dysregulated in response to pathological stimuli, resulting in structural and functional changes in the apical GCX [9–13]. These alterations have been shown to influence endothelial permeability to a wide range of molecules and cells, including lipoproteins, cytokines, immune cells, and cancer cells, thereby underscoring the critical regulatory role of the apical GCX in the development of pathologies [14–19]. Therefore, understanding the structural and compositional response of the apical GCX to physiologically relevant forces has been a key focus of the field [9,20–22].

**Data availability statement:** All image data that supports the results of this study, as well as all novel ImageJ macro-functions and CellProfiler pipeline generated for the analysis of the image data collected, have been stored in the DRYAD database which can be accessed at DIO: https://doi.org/10.5061/dryad.c59zw3rn1. Both authors had access to all data included in the study presented.

**Funding:** This work was funded by startup funding awarded to Prof. Solomon A Mensah by Worcester Polytechnic Institute.

**Competing interests:** The authors have declared that no competing interests exist.

**Abbreviations:** GCX, Glycocalyx; EC, Endothelial cell; GAG, Glycosaminoglycan; FSS, Fluid shear stress; HLMVEC, Human lung microvascular endothelial cell; MEGM, Microvascular endothelial growth medium.

The apical GCX is a dense and complex polysaccharide structure that covers ECs [23,24]. Glycoproteins and proteoglycans are the core proteins of the GCX, the two prominent core protein families are syndecan and glypican [5,25,26]. These negatively charged core proteins bind carbohydrate chains such as heparan sulfate and hyaluronic acid and facilitate interactions with other cell adhesion molecules [27–30]. Glycoproteins typically bind short chains and constitute cell adhesion receptors, integrins, and other specialized receptors embedded in the lipid bilayer [27,31]. Proteoglycans and their glycosaminoglycans (GAGs) tower over smaller glycoproteins forming the dense, grass-like structure of the apical GCX as observed in Reitsma et al. [23,25,32]. GAGs dominate the apical GCX and are long, highly polar, chains of repeating disaccharide units that bind to GAG specific binding sites on proteoglycans [33,34]. GAGs, and their associated proteoglycans, have been at the forefront of GCX research due to their prominent role in apical GCX structure and their ability to function as mechanotransducers through transmembrane connections with the EC cytoskeleton [3,4,9,35,36].

Fluid shear stress (FSS) generated by blood flowing against the endothelium, and its role in regulating the apical GCX, has been extensively investigated as endothelium dysfunction often occurs in areas of low or disturbed FSS caused by vessel geometry [37–40]. Vessel curvature and bifurcations create regional variations in FSS due to the complex and multidirectional flow patterns that develop in these areas [11,41,42]. Endothelial permeability and disease susceptibility have been shown to correlate with fluctuations in the time-averaged FSS over a pulsatile cardiac cycle, as well as with the oscillatory nature of the local shear environment [11,39,43]. The apical GCX, which directly interfaces with circulating blood, has been extensively investigated under these varying shear conditions, revealing numerous mechanistic links between FSS and apical GCX-mediated endothelial mechanotransduction [13,44]. In literature, the syndecan proteoglycan family is often investigated as syndecan proteoglycans bind heparan sulfate, the single most prominent GAG in the apical GCX [25,27]. Syndecan proteins, like some other proteoglycans, possess a membrane spanning domain that facilitates connections with the cytoskeleton, allowing extracellular forces to be transmitted directly to the EC cytoskeleton, instigating mechanotransduction pathways [45,46]. Syndecan-1, abundant in the apical GCX, has been shown to be critical in regulating EC behavior and inflammation in response to FSS [36,46,47]. However, recent evidence has shown that syndecan-4, which is present in the basal GCX, also plays a significant role in the EC response to apical FSS [48–54].

It was demonstrated that syndecan-4 facilitates connections with the cytoskeleton modulating known EC FSS mechanotransduction pathways such as Rho and Yap/Taz, with the absence of syndecan-4 resulting in the inability of ECs to align to the flow vector [48,51]. Syndecan-4 has also been examined in the context of EC cytoskeletal remodeling in response to FSS due to its close association with cytoskeletal proteins critical for cytoskeletal remodeling functions [4,48,50,52,53,55]. Thi et al suggested a "bumper-car" model to describe how basal syndecan-4 is affected by FSS through apical integrins transmitting force through the EC cytoskeleton to basal

syndecan-4 [4]. Building on discussed evidence of syndecan-1 and syndecan-4 and in one of the only in vivo observations of the basal GCX, Stoler-Barak et al observed heparan sulfate GCX content across various mice microvessels [56]. Strikingly, during inflammatory stimulation, the basal heparan sulfate content progressively increased within the basal GCX, while the apical heparan sulfate was concurrently degraded, thereby amplifying the heparan sulfate gradient previously observed between the apical and basal GCX of non-inflamed microvessels [56]. This creates a gradient of heparan sulfate across the endothelium, promoting the retention of chemokines within the basal GCX, underscoring the potential, yet unknown, role of the basal GCX in immune cell trafficking. However, even though structural and functional differences have been noted between the basal and apical GCX in response to FSS, research examining the basal GCX is limited to investigation of syndecan-4 and heparan sulfate. A cartoon depiction of all previously documented apical and basal glycocalyx elements and an in depth discussion of the structural and function role of the basal GCX was presented in our previous review paper [54].

We hypothesize that FSS elicits coordinated responses in both the apical and basal GCX through apical GCX and cytoskeletal mediated mechanotransduction [4,48]. Furthermore we hypothesize the response of the apical and basal GCX will also be dependent upon the magnitude of FSS applied [57]. Here, we examine the presence of both the apical and basal GCX in human lung microvascular endothelial cells (HLMVECs) after exposure to a range of physiologically relevant rates of FSS, including a pathophysiologically low, physiologically normal, and physiologically elevated [11,41,42]. This demonstrates for the first time the basal GCX's response to apical FSS and shows that the expression of the apical and basal GCX is significantly different before and after FSS onset. Finally, our findings reveal that the presence of the basal GCX is influenced by both FSS and exposure duration. This suggests a potential role for the basal GCX in the progression of FSS-induced vascular diseases, highlighting the critical need for more targeted investigations into the basal GCX's response to varying shear stress conditions.

## Results

HLMVECs were confirmed to produce a robust apical GCX in response to physiologically normal FSS by probing the GCX following methods previously used to visualize and quantify the luminal GCX (S1 Fig) [9,32,58]. Following exposure to 10 dynes/cm² of FSS, HLMVECs were fixed as described, and their GCX was labeled without permeabilization to ensure that only components naturally accessible to antibodies were tagged during immunostaining. Full thickness z-stacks were not separated using the apical/ basal image separate process described, instead the entire z-stack collected was sum projected and quantified. Metrics of coverage and integrated intensity were found to significantly increase compared to static conditions (p=0.0014, p=0.0135 respectively), and the levels observed after 30 minutes of exposure (p=0.0002, p=0.0113 respectively) (S1 Fig). This indicates a sustained and overall enhancement in GCX expression in response to prolonged shear stress. Measurements of GCX thickness (μm) correspond with the literature ranges reported (S1 Fig) [5,58,59]. These results align with what has been reported for prominent GCX components in other endothelial cell types, such as HUVECs when using the typically presented method of GCX staining, visualization, and quantification [9,12,60–62]. It is also important to note that when z-stacks collected from non-permeabilized HLMVECs are separated into apical and basal stacks and projected, the basal sum projections display signal only near the cell borders, and no measurable basal GCX thickness is observed, as GCX thickness is quantified only above and below each nuclear body (S1 Fig). This suggests that without permeabilization WGA antibodies are not being adequately delivered to the basal HLMVEC GCX.

Most reports of the endothelial GCX have not included permeabilization during immunofluorescent staining, based on the long-standing assumption that the GCX exists solely on the luminal surface [32]. As shown in prior studies, staining with WGA in the absence of permeabilization labels only the apical GCX of confluent endothelial monolayers [9,37,58]. In contrast, when EC are tested with sucrose and saponin, the basal GCX can be successfully visualized [56]. 2% Sucrose and 0.1% saponin were used for membrane stabilization and permeabilization, respectively, over Triton-X100, a common detergent, as initial dose-dependent studies for basal GCX visualization displayed membrane discontinuities post

permeabilization (S2 Fig). The brightfield images revealed disruptions in the cell membrane, which were further confirmed by GCX staining in the orthogonal views (S2 Fig). These observations validate the membrane disruptions that occurred following permeabilization using Triton-X100. When comparing apical GCX signal intensity, quantified as the GCX signal above the z-stack plane corresponding to the centroid of the nuclei, between non-permeabilized HLMVECs and those permeabilized with 0.1% saponin, no significant differences were observed among HLMVECs exposed to FSS for the same duration (Fig 1G–I). These results indicate that sucrose and saponin treatment preserves GCX integrity without significantly altering the apical GCX signal (Fig 1). Subsequent data are grouped into apical and basal GCX signals, obtained from HLMVECs that were permeabilized after flow exposure and fixation. The collected z-stacks were separated using the described method to independently quantify apical and basal GCX metrics.

After HLMVECs were exposed to 10 dynes/cm$^2$ for 0.5 or 12 hours the apical and basal GCX was tagged, imaged, separated into apical and basal signal, and the apical and basal GCX signal was quantified in terms of integrated intensity, coverage (%), and thickness (µm). When comparing GCX thickness, basal GCX thickness decreased significantly after 30 minutes and 12 hours of exposure (p = 0.0138, p < 0.0001 respectively) (Fig 2G). Apical GCX thickness increased slightly with FSS exposure, however, insignificantly. Changes in GCX thickness can be seen in orthogonal views accompanying sum projections in Fig 2A–F. Apical GCX (Fig 2A–C) coverage decreased significantly after 30 minutes of exposure to 10 dynes/cm$^2$ (p = 0.0205), while both apical GCX coverage and integrated intensity (Fig 2H and Fig 2I, respectively) significantly increased between 30 minutes and 12 hours of exposure (p < 0.0001 and p = 0.0038 respectively). Apical GCX expression metrics show an initial GCX disturbance with FSS onset and recovery with extended exposure, as seen in the decrease in coverage and integrated intensity after 30 minutes, followed by a recovery to near static values after 12 hours of exposure to FSS (Fig 2H–I: green). The basal GCX also displayed this disturbance. Basal GCX (Fig 2D–F) integrated intensity (Fig 2I: purple) was found to significantly decrease after 30 minutes of exposure to 10 dynes/cm$^2$ of FSS, recovering to approximately static levels after 12 hours of exposure (p < 0.0001). Basal GCX coverage (Fig 2H: purple) did not vary significantly. Basal GCX coverage and integrated intensity remained significantly higher than apical GCX measures, while inversely, basal GCX thickness remained significantly lower than apical GCX thickness (Fig 2H–I). These relationships were present in all FSS rates tested, aligning with previous observations of a heparan sulfate gradient across the endothelium (Fig 3 and Fig 4) [56].

In HLMVECs exposed to pathophysiologically low FSS at 0.5 dynes/cm², apical GCX metrics showed a downward trend compared to static conditions (Fig 3G-I: green). However, only apical GCX coverage demonstrated a statistically significant decrease after 12 hours of exposure (p = 0.0282) (Fig 3G-I: green). At the 30-minute time point, we see no significant differences between static and 30-minute basal metrics (Fig 3). After 12 hours, the basal GCX showed a significant decrease in coverage and integrated intensity after 12 hours of exposure from static levels and levels after 30 minutes of exposure (p < 0.0001) (Fig 3H-I: purple). Basal GCX thickness was significantly lower after 12 hours of exposure compared to static (Fig 3G) (p = 0.0106).

HLMVECs exposed to high FSS (30 dynes/cm$^2$) showed a significant increase in apical GCX thickness (µm) after 12 hours of exposure compared to static (p = 0.0035) and 30 minutes (0.0042) (Fig 4G). Oppositely, basal GCX thickness significantly decreased after 12 hours (p = 0.0422) (Fig 4G). Apical GCX coverage and integrated intensity significantly decreased after 30 minutes of exposure followed by recovering to static values after 12 hours of exposure (Fig 4G–I: green). Basal GCX coverage and integrated intensity also decreased significantly after 30 minutes of exposure. However, levels remained lowered after 12 hours of exposure (p < 0.0036) (Fig 4H-I). Basal GCX metrics in HLMVECs exposed to 30 dynes/cm² of FSS exhibit a pattern of GCX disruption similar to that observed in HLMVECs exposed to 10 dynes/cm² (Fig 4H-I, Fig 2H-I). Although basal GCX coverage and integrated intensity show a slight increase after 12 hours compared to 30 minutes of exposure, they remain significantly lower than static values following 12 hours of FSS. In contrast, the apical GCX recovers to static levels with prolonged exposure (Fig 4H-I). 30 dynes/cm$^2$ was chosen as a FSS rate that represents an elevated or borderline pathological FSS value for human microvascular ECs [63–65]. When HLMVECs are exposed to 30 dynes/cm² of FSS (Fig 4), there is a clear disruption in basal GCX expression, whereas the apical GCX

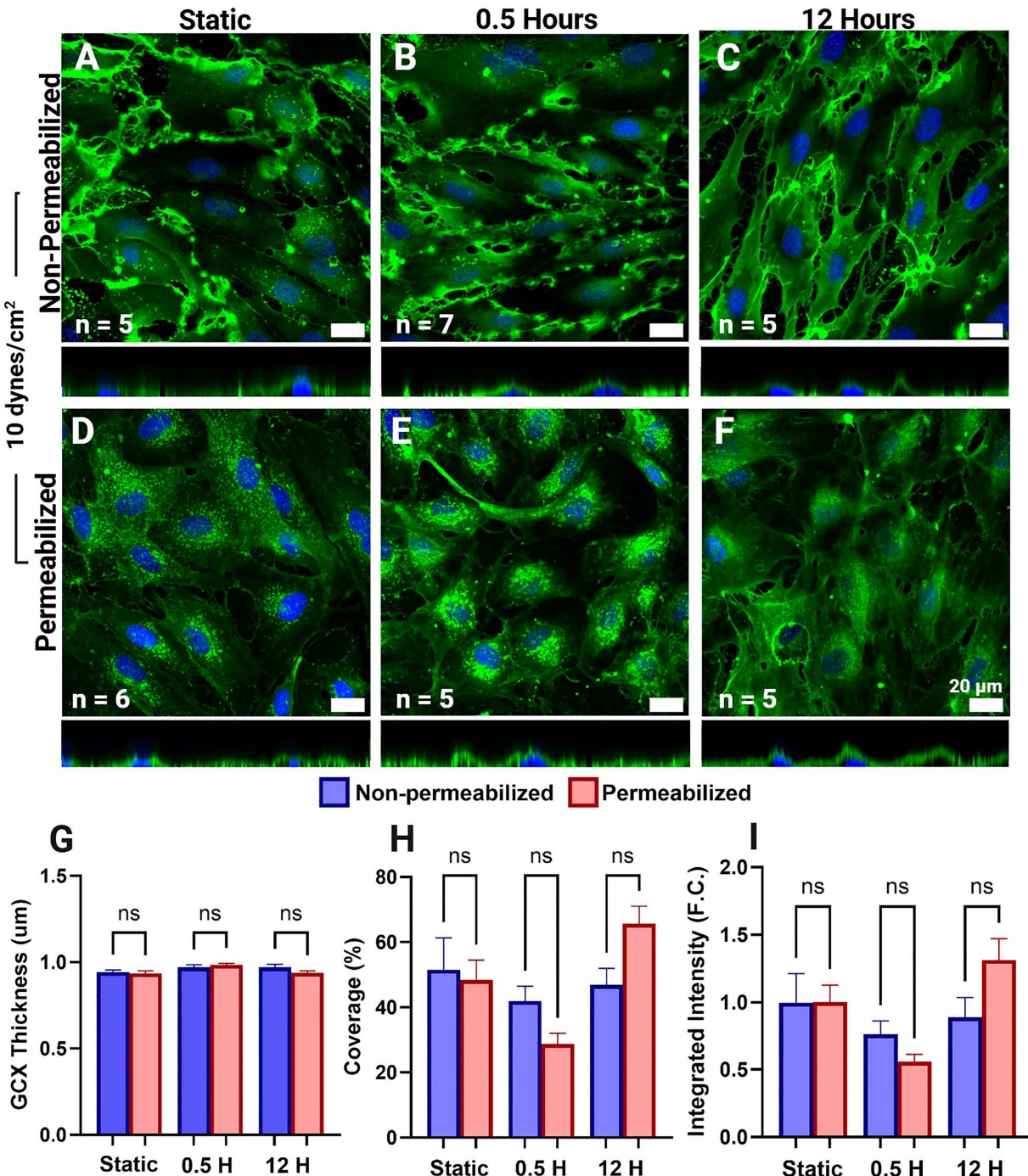

**Fig 1. Comparison of the apical glycocalyx (GCX) signal of permeabilized and non-permeabilized human lung microvascular endothelial cells (HLMVECs) exposed to 10 dynes/cm² of fluid shear stress revealed no statistically significant differences between HLMVECs of the same fluid shear stress (FSS) exposure time.** Sum projections (enface view of A-F) and accompanying orthogonal views (directly below enface view, A-F) of the apical GCX (green) and nuclei (blue) in non-permeabilized (A-C) and permeabilized (D-F) HLMVECs after static culture (A,D), 30 minutes (B,E), and 12 hours (C,F) of exposure to 10 dynes/cm² FSS. No significant differences were found between GCX thickness (µm) (G), coverage (%) (H), and integrated intensity (fold-change (F.C.) (I) of non-permeabilized and permeabilized HLMVECs.

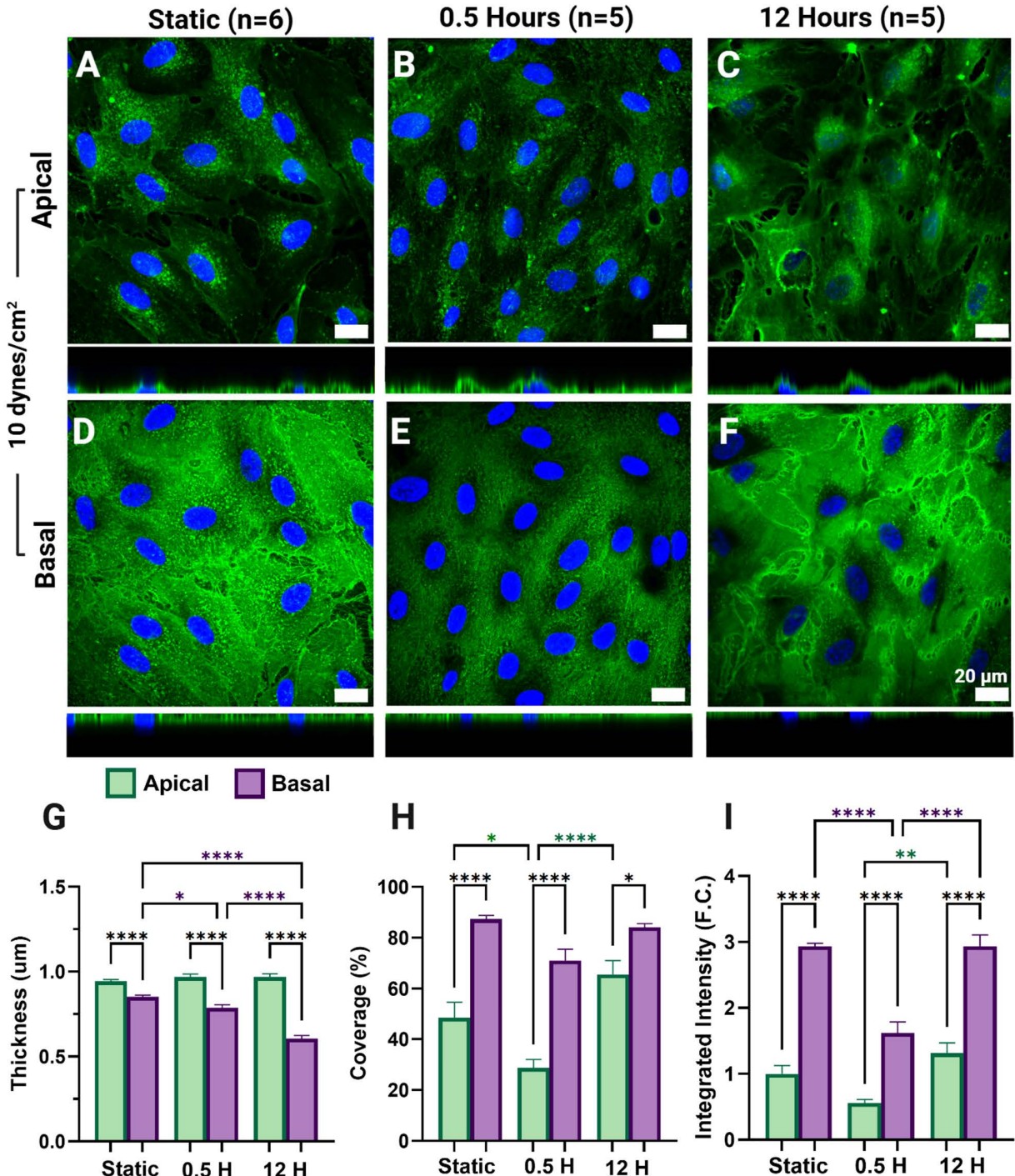

**Fig 2. Apical and basal glycocalyx (GCX) expression in permeabilized human lung microvascular endothelial cells (HLMVECs) after exposure to 10 dynes/cm² of fluid shear stress (FSS).** Sum projection of the apical (A-C) and basal (D-F) GCX (green) and nuclei (blue) (A-C) after static culture (A,D), 30 minutes (B,E), and 12 hours (C,F) of exposure to physiologically normal (10 dynes/cm²) FSS. Apical (green) and basal (purple) GCX thickness (µm) (G), coverage (%) (H), and integrated intensity (fold-change (F.C.) (I) are reported as the mean±standard error of the mean.

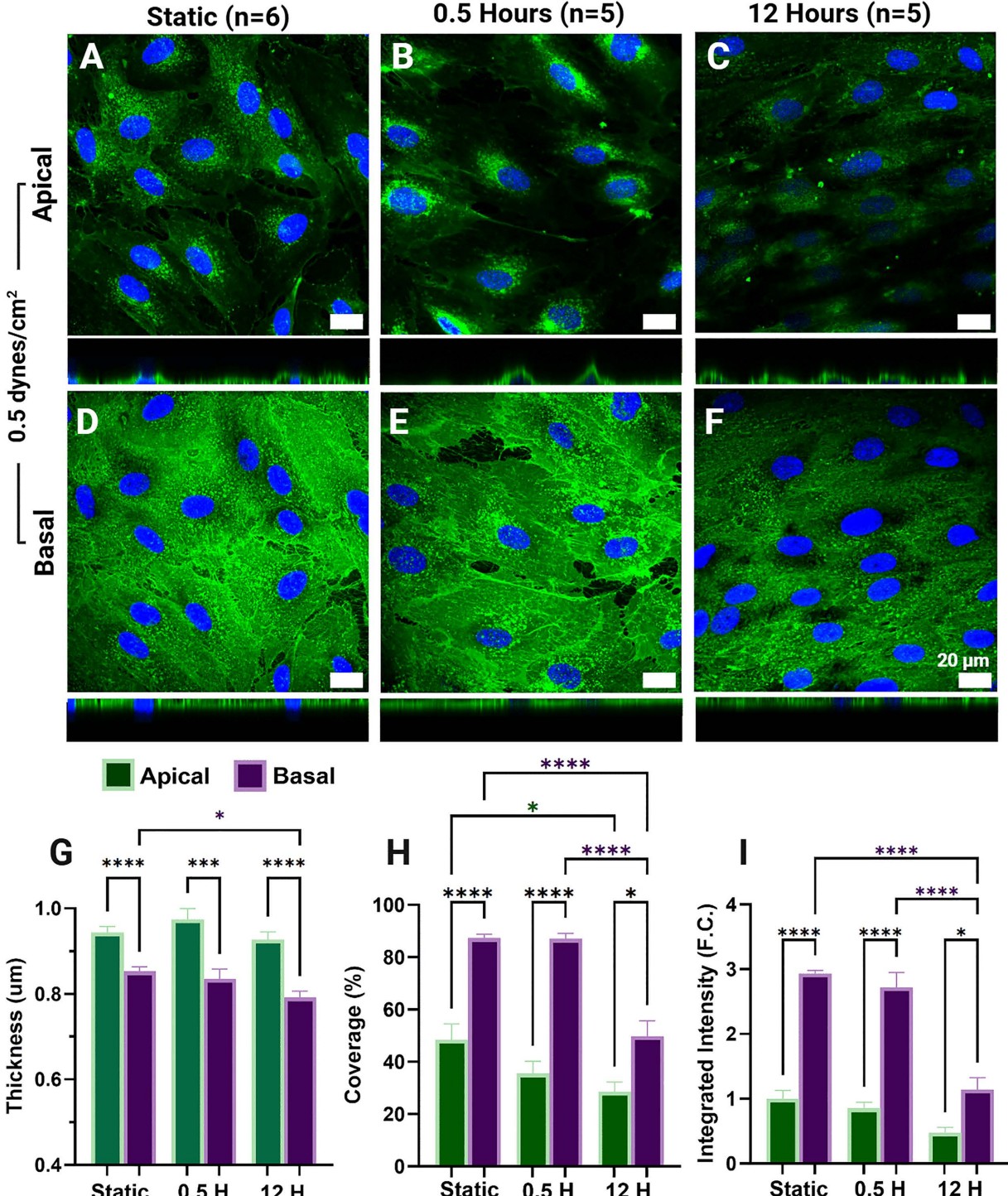

**Fig 3. Apical and basal glycocalyx (GCX) expression in permeabilized human lung microvascular endothelial cells (HLMVECs) after exposure to 0.5 dynes/cm² of fluid shear stress (FSS).** Sum projection of the apical (A-C) and basal (D-F) GCX (green) and nuclei (blue) (A-C) after static culture (A,D), 30 minutes (B,E), and 12 hours (C,F) of exposure to pathophysiologically low (0.5 dynes/cm²) FSS. Apical (green) and basal (purple) GCX thickness (µm) (G), coverage (%) (H), and integrated intensity (fold-change (F.C.) (I) are reported as the mean±standard error of the mean.

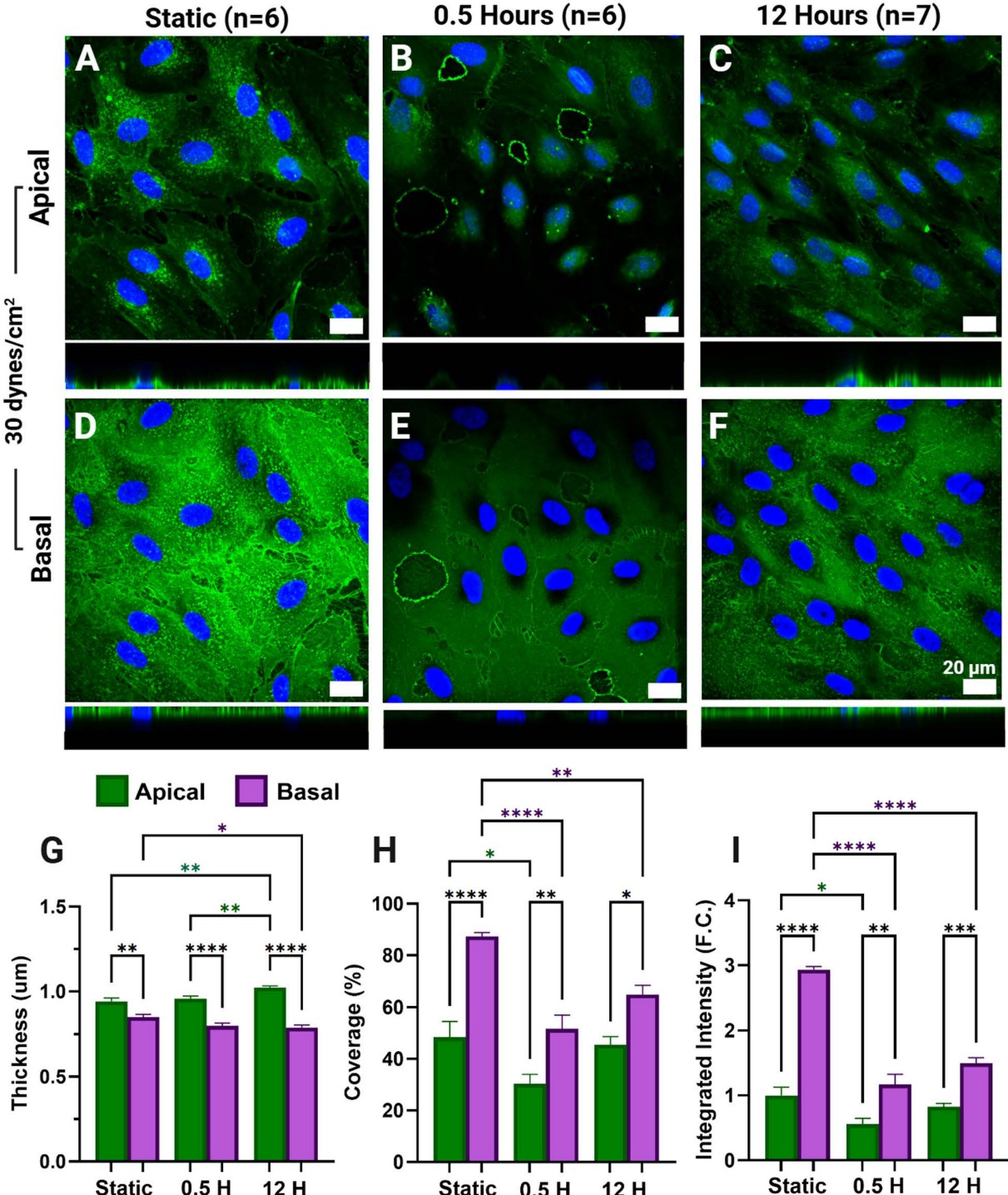

**Fig 4. Apical and basal glycocalyx (GCX) expression in permeabilized human lung microvascular endothelial cells (HLMVECs) after exposure to 30 dynes/cm² of fluid shear stress (FSS).** Sum projection of the apical (A-C) and basal (D-F) GCX (green) and nuclei (blue) (A-C) after static culture (A,D), 30 minutes (B,E), and 12 hours (C,F) of exposure to physiologically high (30 dynes/cm²) FSS. Apical (green) and basal (purple) GCX thickness (μm) (G), coverage (%) (H), and integrated intensity (fold-change (F.C.) (I) are reported as the mean ± standard error of the mean.

integrity appears to recover. In contrast, exposure to low pathological FSS (0.5 dynes/cm²) results in sustained impairment of both apical and basal GCX after 12 hours (Fig 3).

When comparing apical GCX metrics across FSS rates, 0.5 dynes/cm² of FSS significantly reduced apical GCX thickness (μm) compared to apical GCX thickness (μm) after exposure to 30 dynes/cm² of FSS for 12 hours (p = 0.0003). Apical GCX thickness (μm) was never significantly different between 10 dynes/cm² and 0.5 dynes/cm² or 30 dynes/cm² of FSS (Fig 5G). Apical GCX coverage was significantly higher in HLMVECs exposed to 10 dynes/cm² compared to those exposed to 30 dynes/cm², and apical GCX integrated intensity was significantly higher in those exposed to 10 dynes/cm² compared to the low (p < 0.0001) and high (p = 0.0057) FSS tested (Fig 5H-I).

Basal GCX thickness (μm) was significantly affected by FSS rate after 12 hours of exposure (Fig 5J). After 12 hours of exposure, basal GCX thickness (μm) was significantly greater in HLMVECs exposed to 30 dynes/cm² and 0.5 dynes/cm² of FSS compared to those exposed to 10 dynes/cm² (Fig 5J). Basal GCX expression metrics show that after 30 minutes, basal GCX coverage and integrated intensity are highest in HLMVECs exposed to 0.5 dynes/cm², followed by HLMVECs exposed to 10 dynes/cm² and subsequently those exposed to 30 dynes/cm². This suggests that the intensity of shear stress may directly influence the extent of basal GCX disruption during the onset of FSS. After 12 hours of exposure, basal integrated intensity and coverage are significantly elevated in HLMVECs exposed to 10 dynes/cm² of FSS. HLMVEC integrated intensity was not significantly different between HLMVECs exposed to 0.5 dynes/cm² and 30 dynes/cm² of FSS; however, HLMVEC basal GCX coverage was significantly lower in HLMVECs exposed to 0.5 dynes/cm² of FSS compared to 30 dynes/cm² (Fig 5L). Cumulative basal GCX metrics suggest the basal GCX in HLMVECs experiences disruptions when exposed to pathologically low and elevated FSS rates, reflecting similar dysregulation of GCX shedding and production as documented in the apical GCX in response to atheroprone FSS.

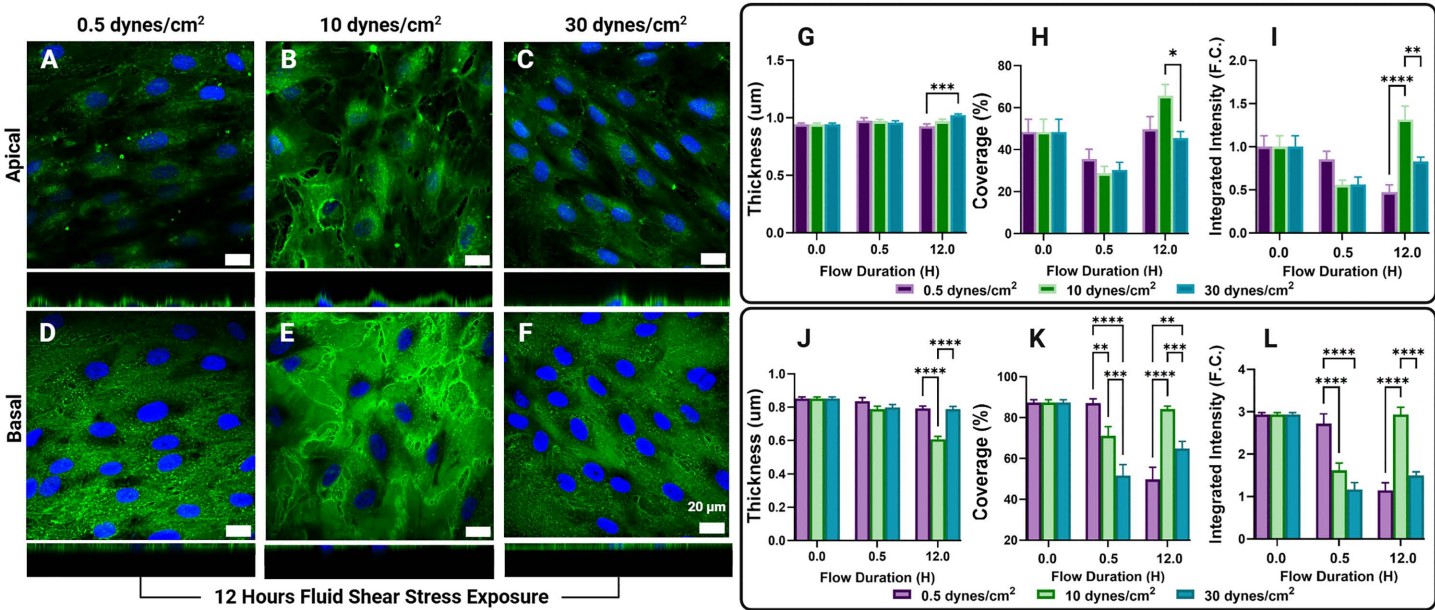

**Fig 5. Comparison of apical and basal glycocalyx (GCX) expression in permeabilized human lung microvascular endothelial cells (HLMVECs) across fluid shear stress (FSS) magnitudes.** Sum projections (top image) and accompanying orthogonal view (bottom image) of the apical (A-C) and basal (D-F) GCX (green) and nuclei (blue) after 12 hours of exposure to 10 dynes/cm² (physiologically normal)(B,E), 0.5 dynes/cm² (pathophysiologically low)(A,D),and 30 dynes/cm² (elevated)(C,F)FSS. Apical GCX thickness (μm), coverage (%), and integrated intensity (fold-change (F.C.) (G-I) are compared between FSS rates after 30 minutes and 12 hours of exposure. FSS comparisons are repeated for basal GCX thickness (μm), coverage (%), and integrated intensity (fold-change (F.C.) (J-L).

## Discussion

Here, we investigate the modulation of both the apical and basal GCX in response to apical FSS. We leveraged fixation and permeabilization methods previously presented by Stoler-Barak et. al to deliver immunofluorescent antibodies to the basal GCX and laser scanning confocal microscopy to collect the 3D signature of the HLMVEC GCX. Our quantitative image separation tool, based on methods previously presented to probe topological gradients in EC expression patterns, allows us to separate and quantify the apical GCX and basal GCX separately as shown in Fig 6 [9,56]. We demonstrated that these methods do not disrupt apical GCX signal (Fig 1) allowing us to probe the temporal response of the apical and basal GCX to FSS instigation (Fig 2, Fig 3, Fig 4) and compare their expression after exposure to physiologically relevant FSS rates (Fig 5). The shear rates of 0.5 dynes/cm$^2$, 10 dynes/cm$^2$, and 30 dynes/cm$^2$ were chosen to replicate physiologically relevant FSS rates observed by microvascular ECs in areas of curvature or bifurcation [11,41,42]. ECs exposed to 10 dynes/cm$^2$ have been shown to produce low levels of inflammatory markers and exhibit antithrombotic phenotypes with prolonged exposure in microvascular ECs [64,66–68]. Therefore, 10 dynes/cm$^2$ was chosen to represent a physiologically normal FSS rate. Literature suggests that even with physiologically normal FSS rates, FSS onset initially leads to GCX degradation [9,15,22,69,70]. Evidence of GCX degradation with FSS onset has been documented in vivo through evaluating the levels of GCX components such as heparan sulfate and syndecan-1 in circulating blood levels after reperfusion events [15,69,71]. These findings have been replicated in vitro, showing that the onset of FSS alone disrupts the GCX at early time points [9,22]. Additionally, these studies demonstrate that prolonged exposure to FSS leads to the formation of a robust GCX [9,12,22]. This characteristic behavior, previously only observed in the apical GCX, was replicated in our findings of both the apical and basal GCX (Fig 2I-H). The reason for apical GCX shedding in response to flow onset has been investigated in the context of reperfusion ischemia [69]. Strong evidence shows that many prominent GCX components are shed in response to reperfusion, however, the mechanism is unknown, and evidence of GCX shedding has been shown to precede the production of inflammatory cytokines and leukocyte adhesion molecules [72]. It has been proposed that the initial release of GCX components may be triggered by the presence of other circulating molecules [69,72]. Circulating peptides and enzymes such as atrial natriuretic peptides and heparinase have been observed to increase in circulation in cases of ischemia and reperfusion injury. These molecules, often referred to as 'shedases,' mediate rapid GCX degradation by either promoting syndecan-1 ectodomain shedding or by enzymatically cleaving heparan sulfate chains from their core proteins (heparinase), leading to acute GCX loss and endothelial dysfunction [69,73]. However, this does not account for the observed decrease in basal GCX expression, as circulating shedases alone cannot explain this effect [19].

We hypothesize that the basal GCX shedding observed after 30 minutes of exposure is resultant of traction forces generated in the basal EC due to apical FSS. Weinbaum et al. suggested that regardless of the state of the apical GCX, the basal EC will be subjected to the same resultant force from the apical FSS [74]. Others have investigated this idea in attempts to understand force transduction throughout the EC [57,75,76]. Thi et al. developed a theoretical bumper car model to reflect their findings of the interplay between the GCX, integrin clusters, and EC cytoskeletal elements in response to FSS [4]. Their model suggests that the onset of physiologically normal FSS produces a resultant torque transmitted through the actin cortical web to intercellular adherens junctions and dense peripheral actin bands (DPABs). This results in a disruption of the DPABs, and the basal stress fibers linked to them. DPABs are recruited to the basal EC to form basal focal adhesions stabilizing the EC. Meanwhile, stress fibers that connect the apical and basal membranes shown to colocalize with apical and basal GCX elements as well as integrin clusters, help transmit the shear force experienced by the apical EC membrane to the basal membrane [4]. We hypothesize that the force initially transmitted through stress fibers generates tension in basal GCX core proteins and GAGs bound to the extracellular matrix, leading to GCX shedding [4,50]. This process mirrors the apical GCX response and may contribute to increased permeability observed following reperfusion ischemia [9,69].

With prolonged exposure to uniform FSS, ECs are able to stabilize through the reformation of DPABs and junctional complex and reorganization of focal adhesions [4]. Here, we observe both apical and basal expression metrics return to

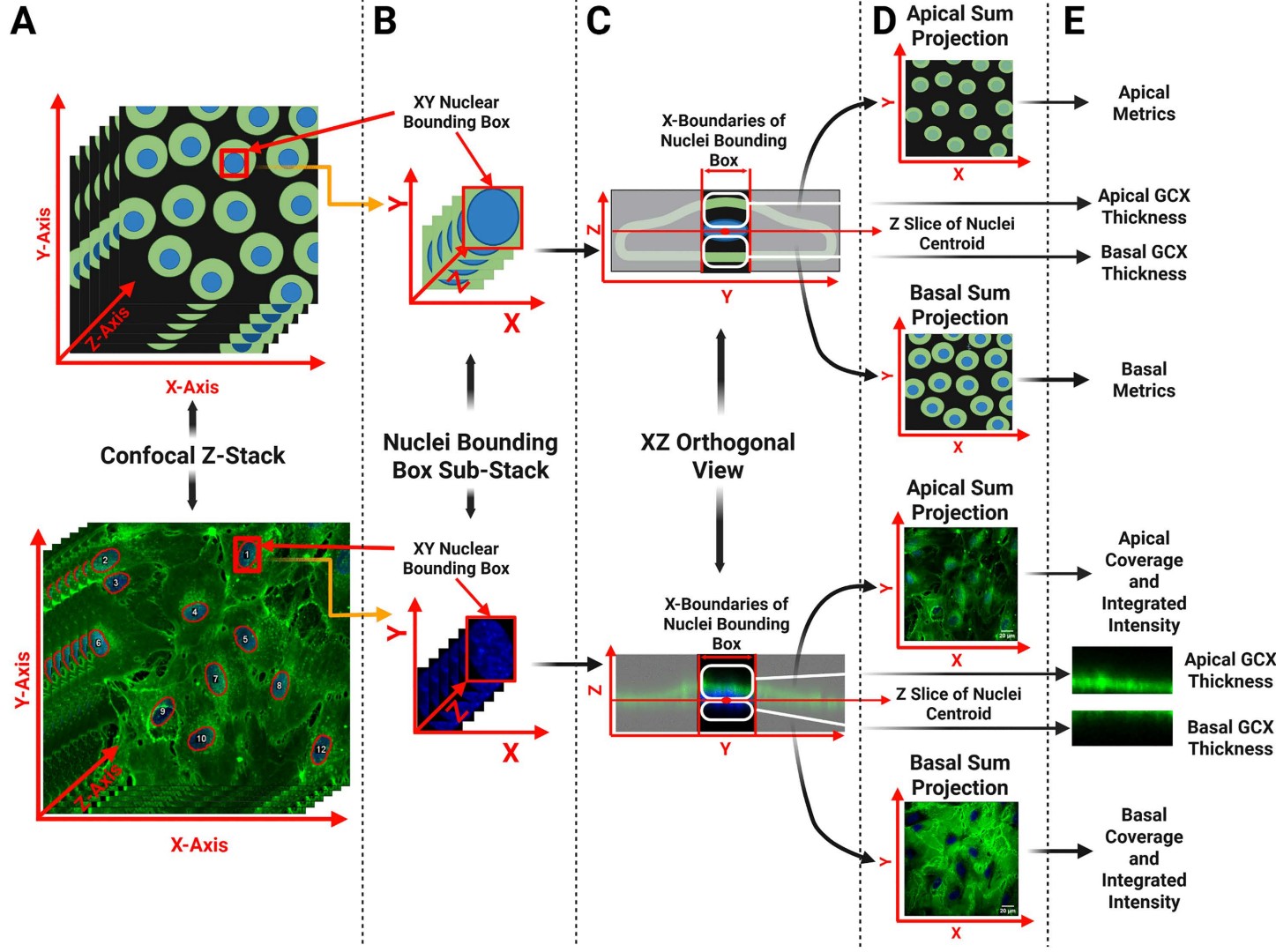

**Fig 6. Z-stack processing to separate full thickness z-stack into apical and basal glycocalyx (GCX) sum projections and orthogonal views for quantification.** The upper progression of images represents an animated version of the step of the image analysis pathway below shown as a real image from the pipeline. All nuclei completely contained within the field of view (FOV) of the full thickness confocal Z-stack (A) are identified through thresholding a basic maximum projection generated of only the nuclei channel of the Z-stack. The identified nuclei bodies are overlaid with the confocal Z-stack as shown in the lower confocal Z-stack (A). A minimum bounding box is drawn around each nucleus and used to crop the original Z-stack into sub-stacks of the GCX, and nuclei signal as shown in panel B. From each sub-stack, an orthogonal view is generated (C) in both the GCX and nucleus channel shown merged here (C). Within the nuclei orthogonal view, the Z-centroid of the nucleus is identified (C). The GCX orthogonal view of the sub-stack is then split into two orthogonal views, and apical and basal GCX orthogonal view, as shown in panel E. This process is repeated on each nucleus identified initially (A) generating a list of nucleus Z-centroid locations and apical and basal GCX orthogonal views. Measurements of apical and basal GCX thickness are made only in the apical and basal orthogonal views extracted from this process. The average Z location of all nuclei centroids is used to split the Z-stack at the Z-stack slice that corresponds to the average center of the nuclei heights into an apical and a basal sub-stack. These sub-stacks are used to generate sum-projections of the GCX signal (D) for analysis of GCX expression metrics of coverage and integrated intensity (E). Created in https://BioRender.com.

levels similar to their static expression after 12 hours, aligning with literature reports of apical GCX recovery [9,70]. To our knowledge, this is the first study examining the isolated effects of apical FSS on the basal GCX. Therefore, the observation of basal GCX modulation and specifically recovery with FSS was unknown, but unsurprising, as predominantly basal

glycoproteins have been found to play key roles in the EC response to FSS [48]. Stoler-Barak et al. demonstrated that the basolateral EC had significantly increased heparan sulfate content compared to the apical GCX with improved resolution imaging of mice ECs [56]. Therefore, the restoration of the basal GCX signal with FSS exposure was anticipated. Also, at all-time points and FSS rates tested, expression metrics of coverage and integrated intensity were significantly higher in the basal GCX compared to the apical GCX (Fig 2, Fig 3, Fig 4). The increased expression of the basal GCX compared to the apical may suggest that EC barrier functions previously solely attributed to the apical GCX may be a function of the basal GCX as well. Stoler-Barak et al. findings even suggest that the apical – basal gradient generated by GAGs may be important for driving these barrier functions, much like LDL gradients driving further deposition in the basement membrane of ECs [14,56].

Pathologically low FSS rates, such as the 0.5 dynes/cm$^2$ tested here, replicate shear stress rates found at vessel bifurcations and surrounding atherosclerotic plaques or vessel thrombosis [77–79]. Microvascular ECs exposed to FSS rates of 0.5 dynes/cm$^2$ have been shown to produce elevated levels of inflammatory markers, indicating a pathological state or response to low FSS[75,80–82]. Literature reports indicate that when the GCX is exposed to pathophysiologically low FSS, its expression decreases compared to physiologically normal FSS [13,83,84]. This reduction is demonstrated through immunofluorescence staining, as well as mRNA and protein expression analyses of GCX core proteins and GAGs [13,83,84] Our apical GCX results reflect these findings with apical GCX coverage, integrated intensity, and thickness trending up after FSS onset (Fig 3, Fig 5). The trend of decreasing expression is replicated in the basal GCX however, only after prolonged exposure. The combined findings of Thi et al. and Weinbaum et al., as previously discussed, suggest the existence of a torque threshold. If this threshold is not met due to low FSS acting on the apical GCX, it does not lead to DPAB formation, stress fiber fragmentation, or basal focal adhesion remodeling [4,74]. We believe that these low resultant forces do not exceed the force threshold necessary to promote basal GCX shedding with apical FSS initiation rather, the delay observed in the basal GCX degradation could be due to excited inflammatory pathways resultant of the low FSS rather than the reaction force generated in the basal [81–83,85]. Apical GCX disruption due to low shear stress has been shown to increase EC permeability [83]. These low-shear environments also promote EC interactions with various circulating molecules and cells, such as cancer cells, immune cells, LDLs, and coagulation factors, due to the reduced EC barrier function and diminished shear stress at the vessel wall [11,39,43,58,77]. A more limited set of evidence has explored how these cells and molecules are trafficked through the vessel once bound to the EC surface. The importance of junctional strength has been strongly emphasized along with the basal EC environment for trans endothelial migration [86–88]. Coupled with primary evidence previously discussed regarding how heparan sulfate gradients are generated across the EC in inflamed environments resulting in entrapment of chemokines in the basal ECs. The decrease in the bulk presence of both apical and basal GCX and the maintenance of the apical, basal GCX signal gradient observed here after prolonged exposure to physiologically low FSS may shed light on how EC permeability is dependent upon both apical and basal GCX presence and affected by pathological flow promoting migratory processes and the deposition of other small molecules (Fig 3, Fig 5) [89–92]. Suggesting the role of the basal GCX, and its dynamic elements, in modulating EC permeability should be further explored as endothelium hyperpermeability is relevant in the development of many pathologies [19,69,83,93,94].

Numerous studies have demonstrated the role of FSS in the development of atherosclerosis, while elevated FSS has been strongly associated with high-risk plaque characteristics [83,95–97]. Elevated FSS rates such as 30 dynes/cm$^2$ have also been observed in fibrotic or diseased conditions; however, in reports isolating the effect of elevated FSS on ECs, a variety of both normal and pathological EC phenotypes have been observed [14,78,98]. It remains unclear whether elevated FSS observed with a variety of vascular diseases is causal or resultant of the disease progression [79]. In HLMVECs exposed to 30 dynes/cm², the apical expression pattern resembles that observed in HLMVECs exposed to 10 dynes/cm². Expression metrics initially decrease but recover to near-static levels after 12 hours. However, basal expression metrics of coverage and integrated intensity are significantly lower in HLMVECs exposed to 30 dynes/cm$^2$ compared

to those exposed to 10 dynes/cm$^2$ (Fig 5K-L). Basal expression metrics suggest that the basal GCX did not recover as significantly as the apical GCX with prolonged exposure (Fig 5J-L). This is evident in the substantial decrease compared to static levels, whereas the apical GCX expression and levels returned to baseline after 12 hours of exposure. This may suggest that the basal GCX is more affected by elevated FSS rates than the apical.

Due to the flat morphology of confluent ECs plated on glass slides, even with high-resolution imaging, it becomes impossible to discern the apical and basal membrane outside of the membrane directly over and under the EC nucleus. This is easily seen in the orthogonal views accompanying apical and basal sum projections in Fig 2. It is for this reason that the apical and basal GCX signal was divided at the centroid of the nucleus for expression measurements following reports of others probing topological expression in EC and the GCX specifically [9,56]. This choice was supported by apical GCX expression metrics following patterns presented in the literature [9,12,22,56]. Apical and basal GCX thickness measurements were specific to the apical and basal GCX as GCX thickness was only probed directly above and below the nucleus. This allowed for quantitative measures specifically comparing the apical and basal membrane GCX. Following literature reports, apical GCX thickness trended up with prolonged exposure to physiologically normal FSS and exhibited a significant decrease in apical GCX thickness low and high FSS rates [70,97,99]. Basal GCX thickness values have not previously been directly presented in the literature to our knowledge. Here, we observed that under physiologically normal FSS, the basal GCX thickness decreases with prolonged exposure below apical thickness levels. This result contradicts the findings of Stoler-Barak et al., which reported that basal heparan sulfate thickness is greater than apical heparan sulfate thickness [56]. However, the methodologies used to probe the GCX were significantly different, and Stoler-Barak et al. specifically examined only heparan sulfate in mice ECs [56]. A surprising relationship between basal GCX thickness and inflammatory stimuli was found both here and in Stoler-Barak et al.. They found that in inflamed micro vessels apical heparan sulfate was abolished while basal heparan sulfate was enriched compared to non-heparinase-III treated mice [56]. Here, we observed that after 12 hours of exposure, basal GCX thickness was significantly higher in HLMVECs exposed to 0.5 dynes/cm$^2$ and 30 dynes/cm$^2$ compared to those exposed to 10 dynes/cm$^2$. Both results may suggest that inflammatory apical stimuli promote basal enrichment, possibly contributing to EC barrier functions.

The in vitro flow conditions used in this study were designed to mimic the flow profiles typical of the vasculature. By examining EC and GCX responses under these shear environments, our findings provide insights into how local hemodynamic forces drive endothelial GCX response and gradients across the endothelium. Overall, when comparing apical and basal GCX metrics between groups of different FSS exposure we observed that apical and basal GCX metrics are significantly decreased in HLMVECs exposed to modulated FSS rates (0.5 and 30 dynes/cm$^2$) compared to HLMVECs exposed to 10 dynes/cm$^2$ (Fig 5). Relating to observations of increased endothelium permeability previously demonstrated in regions of modulated wall shear stress [11,39,43]. As GCX presence or abundance is correlated to barrier functions, our data suggest that the role of the basal GCX should be considered when investigating GCX and FSS dependent modulation of endothelium permeability functions [19,62,97,100]. Emerging evidence suggests that apical FSS can influence the basal GCX through cytoskeleton-mediated mechanotransduction. In the bumper-car model proposed by Thi et al. , deformation of HS proteoglycans (syndecan-1 and syndecan-4) under shear stress initiates intracellular signaling that extends across the endothelial cell to the basal surface [4]. Such coupling may explain how changes in apical shear can regulate basal GCX remodeling and basement membrane interactions [4]. We hypothesize that changes in signaling domain tension due to the differences in FSS experienced by our experimental groups, previously shown to regulate GAG transcription and apical GCX presence, may also regulate basal GCX presence and the difference observed when comparing groups across FSS magnitude.

In summary, using optimized fixation and permeabilization methods, we were able to visualize and quantify the basal GCX signal under physiologically relevant shear conditions, revealing time- and shear-dependent modulation not previously reported. These results suggest that the basal GCX is sensitive to apical FSS, exhibiting distinct expression patterns compared to the apical GCX and showing higher overall expression. Our results extend existing models of GCX

mechanotransduction by suggesting that apical shear forces may be transduced through cytoskeletal coupling mediated signaling pathways to regulate basal GCX organization and presence [4]. This dual-surface analysis advances current understanding of endothelial mechanobiology by linking apical flow sensing to basal matrix interactions, thereby providing a more complete picture of GCX-mediated endothelial function. This may suggest that for disease with primary development in the vascular bed or relation to vascular permeability, the basal GCX and its effect on EC barrier function and mechanotransduction should be considered along with apical function [97,100]. This demonstrates a major gap in our understanding of EC mechanotransduction. Our recent review paper details research objectives necessary for addressing major gaps in our understanding of basal GCX function, specifically in the context of the ECs response to FSS [54]. These include probing specific GCX elements such as heparan sulfate, hyaluronic acid, and chondroitin sulfate to understand their presence in the basal GCX. The exact structural composition of basal proteoglycans and molecular weights of basal GCX GAGs should be investigated and compared to apical counterparts. Not only should the presence of individual GAGs and their core protein response to FSS be observed but the dependence of the basal GCX elements modulation in response to FSS in the presence of the apical GCX should be probed. For example, when evaluating the endothelial response to FSS, it would be important to determine whether the observed effects are influenced by components located on the apical surface or by cytoskeletal dynamics. This could be examined by enzymatically removing apical surface constituents or by inhibiting actin polymerization. Such experiments would refine current models of endothelial FSS mechanotransduction and deepen our understanding of how the GCX regulates vascular permeability. Future studies should extend the duration of FSS exposure to enable investigation of steady-state expression levels of GAGs and proteoglycans, providing additional evidence for their physiological relevance in vivo. Moreover, functional analyses should examine how the presence and modulation of basal GCX components influence trans endothelial migration of immune and cancer cells, as well as the transport of macromolecules such as LDLs and cytokines that play key roles in disease progression. Similar to studies on apical GCX function, research on basal GCX function should also characterize basal GCX modulation and function in the EC response to other physiologically relevant forces in the basement membrane such as interstitial or transmural flow and substrate stiffness and composition. Ultimately studies should probe the basal GCX in vivo to examine basal GCX presence in both large and microvascular networks utilizing healthy and diseased models to shed more light on the basal GCX in relation to endothelium function.

## Materials and methods

### Cell culture and shear application

HLMVEC (Cell Applications Inc.) were cultured and maintained in microvascular endothelial growth medium (MEGM) (Cell Applications Inc.). Cells passages 5–7 were plated on fibronectin coated (10 µg/ml) glass slides (ThermoFisher Scientific) at a density of 200,000 cells/mL and allowed to reach confluency over 72–84 hours to ensure full maturation of the GCX. This culture time allows for robust monolayer formation and aligns with previously presented experimental protocols [12,101]. A confluent HLMVEC slide was placed on the base plate of a Glycotech parallel plate flow chamber, with a gasket positioned on top to regulate the chamber's thickness. The upper plate contained both the inlet and outlet ports, allowing a peristaltic pump to continuously circulate MEGM through a closed flow loop [37,102]. Shear rates of 0.5 dyne/cm$^2$, a pathophysiological low shear rate [64,103], 10 dynes/cm$^2$, a physiological normal shear rate [9,22,62,101], and 30 dynes/cm$^2$, an elevated shear rate for microvascular ECs [64,103] were tested. During shear exposure the flow system was kept at 37°C and 5% $CO_2$ humidified air incubator.

### Immunofluorescence staining

After flow exposure, HLMVEC were briefly washed with 3% bovine serum albumin (BSA) in PBS before fixation with 4% paraformaldehyde and 2% sucrose in PBS for 10 minutes. The fixation solution was supplemented with 2% sucrose for membrane stabilization during permeabilization [56]. A blocking solution of 3% BSA was supplemented with 0.1% (wt/v)

saponin for permeabilization to deliver immunofluorescence antibodies to the basal surface of the HLMVECs. Saponin was omitted for non-permeabilized samples presented in Fig 1 and S1 Fig. Samples were allowed to incubate in primary solution containing biotinylated wheat germ agglutinin (WGA) (1:500, Vector) in 1% BSA in PBS overnight. WGA was utilized as it binds broadly to sialic acid- and GlcNAc-containing moieties within the GCX [32,37,56]. Negative controls were generated by omitting WGA in the primary solution. The secondary detection solution contained Streptavidin Alexa Fluor 488 (1:1000, Invitrogen) in 1% BSA in PBS. Samples were mounted with Fluoromount-G with DAPI mounting media for immunofluorescent nuclei tagging (Invitrogen).

## Confocal microscopy quantitative apical/basal analysis

Samples were imaged on a Leica STELLARIS 8 laser scanning confocal microscope using a plan-apochromat 63 x 1.40 NA oil objective. At a randomly selected area within the flow path, a full thickness z-stack was collected with a slice thickness of 0.1 µm and field of view (FOV) of 184.7 x 184.7 µm. Each Z-stack was captured as an 8-bit image series with resolution of 1024 x 1024 pixels. Image stacks were first analyzed in ImageJ (1.54i, NIH) to generate sum projections of the apical and basal GCX signal for further analysis.

## Quantitative apical/basal analysis

To generate sum projections of the apical and basal GCX signal, the apical and basal aspects of the z-stack were distinguished by the average Z-centroid location of all nuclei completely contained within the image FOV, as shown in Fig 6. Initially both the XZ and YZ orthogonal view of each nucleus was analyzed to measure the Z-centroid of each nucleus across conditions. We typically captured ~20 nuclei per field of view and found no significant difference in Z-centroid measurements between XZ and YZ views for cells exposed to 10 dynes/cm² FSS. This confirms that the nuclear centroid measurement, and therefore our z-slice segmentation was not affected by cell elongation or polarity changes following flow exposure. To improve efficiency without compromising accuracy, subsequent analyses used only the XZ orthogonal view, which aligns with the flow direction and significantly reduced processing time. Therefore, the Z-stack slice that corresponds to the average height of all nuclei centroids, as measured in the XZ plane, was used to divide the original stack into apical and basal sub-stacks following methods previously presented for EC vertical spatial distribution analysis as seen and described in Fig 6. These sub-stacks were used to generate sum-projections of the apical and basal GCX signal, following methods previously presented [9,56]. A Gaussian filter with a sigma value of 1 was applied to smooth apical and basal GCX projections prior to analysis.

The apical and basal sum-projections were analyzed in CellProfiler (4.2.6, Broad Institute, Inc.) to measure GCX coverage and integrated intensity as a general measure of GCX expression in both the apical and basal cell [104]. Coverage was defined as the percentage of pixels within the image FOV with an intensity greater than the background intensity. Background intensity was determined to be 6% and 8% of the normalized distribution of pixel intensities in the apical and basal GCX, respectively, corresponding to the average integrated intensity measured from apical and basal negative control samples. The integrated intensity was calculated as the sum of pixel intensities determined to be positive for GCX coverage. Here, all metrics of integrated intensity have been normalized around the mean static apical GCX integrated intensity value.

GCX thickness (µm) was measured in XZ orthogonal view images generated at the XY centroid of each nucleus completely contained within the image field of view. Here orthogonal views were generated in ImageJ through the same custom ImageJ macro-function that generates the apical and basal sum projections. Due to the morphology of ECs plated on hard substrates, GCX signals not directly above and below the physical nuclei body cannot be separated into apical and basal signals confidently. Therefore, only the signal within the X-dimension bounding box limits of the nucleus was analyzed, as shown in S3 Fig. GCX thickness (µm) measurements were extracted in Cell Profiler by extracting the average Z height of the GCX signal exceeding background intensity within the X-bounding box of the XZ orthogonal views

generated. The average value of apical and basal GCX thickness (µm) was reported by averaging the values extracted for each nucleus within the FOV to represent each image.

## Statistics and data representation

Each experimental sample is represented as one set of apical and basal metrics (integrated intensity, coverage, and thickness) such that the metric can be compared between apical and basal, across exposure times (apical 0.5-hour metrics vs. apical 12-hour metrics), and with other apical and basal metrics after exposure to a different FSS rate (basal 12-hour metrics: 10 dynes/cm$^2$ vs 0.5 dynes/cm$^2$). All qualitative data is represented by the mean of the group with variance presented as the standard error of the mean generated by at least five independent experiments (numerical replicates subsequently listed for each experimental group in Results). Multi-comparison one-way ANOVA tests with Brown-Forsythe and Welch parameters were performed to determine statistical significance differences between apical and basal means at each timepoint, and across apical and basal means independently when comparing GCX expression metrics over FSS exposure time within one shear rate. A two-way grouped ANOVA test was run to determine the effects of FSS rate and time on each GCX metric. P-values were considered significant if less than 0.05. All resultant P-values can be found in S1 Table. All statics and data plots were generated using GraphPad Prism 10 (version 6.04 for Windows, GraphPad Software, www.graphpad.com).

## Supporting information

**S1 Fig. Effect of physiologically normal (10 dynes/cm$^2$) fluid shear stress (FSS) on non-permeabilized human lung microvascular endothelial cells (HLMVECs) glycocalyx (GCX) expression.** Sum projection and accompanying orthogonal view of the non-permeabilized HLMVEC GCX after static culture (A) and 0.5 H (B), and 12 H (C) of exposure to 10 dynes/cm$^2$ of FSS. GCX thickness (um)(D), coverage (%)(E), and integrated intensity (fold-change (F.C.)(F) in HLMVECs after static culture, 30 minutes, and 12 hours of exposure to physiologically normal (10 dynes/cm$^2$) FSS exposure. WGA staining was performed on non-permeabilized post-fix HLMVECs. Here, no apical/ basal image separation was performed, and all slices of the full thickness z-stack are summed to a single sum projection and the metrics presented were quantified to ensure the bulk non-permeabilized expression of the HLMVEC GCX displays FSS dependent expression. GCX thickness was analyzed across the entire Z-stacks YZ orthogonal view of the z-stacks collected.
(TIF)

**S2 Fig. Permeabilization effects by permeabilization treatment method.** Brightfield images of (A) the HLMVEC monolayer post-fix, (B) after treatment with 0.1% (wt/v) saponin for 1 hour during blocking, (C) and after completion of the immunofluorescence staining process prior to sample mounting. HLMVEC monolayer after 3, 5-minute treatments of 0.025% (D), 0.015% (E), and 0.01% Triton-100 in PBS post-fix. Sum projection and accompanying orthogonal view of glycocalyx signal of non-permeabilized HLMVEC (G), HLMVEC permeabilized with 0.1% (K), 0.05% (L), and 0.025% Triton-100 (J), and 0.1% (K), 0.075% (L), and 0.05% saponin (J) after exposure to 10 dynes/cm$^2$ of fluid shear stress for 30 minutes.
(TIF)

**S1 Table. Detailed statistical results for all results presented.** The significance level (Sig), P-values (P), and degree of freedom (DF) are reported for each individual comparison generated within the Anova results presented in Figure 1 – Figure 5 (A – E). The overall Anova p-value (P) along with the sum of the treatment and residual degrees of freedom (F) for each glycocalyx (GCX) metric (coverage, integrated intensity, and thickness) are reported. Significance level correspond to the following ranges: (ns: p =>0.005), (*: p<0.05), (**: p<0.01), (***: p<0.001). Naming samples includes the shear stress exposure time in hours (H) and rate in dynes/cm$^2$ (D) therefore the experimental group exposed to 10 dynes/cm$^2$ of fluid shear stress for 0.5 hours would be labeled as 0.5H10D. Non-permeabilized (NP), permeabilized (P).
(DOCX)

## Author contributions

**Conceptualization:** Solomon A. Mensah.

**Data curation:** Zoe Vittum.

**Formal analysis:** Zoe Vittum.

**Funding acquisition:** Solomon A. Mensah.

**Investigation:** Zoe Vittum, Solomon A. Mensah.

**Methodology:** Zoe Vittum.

**Project administration:** Solomon A. Mensah.

**Resources:** Solomon A. Mensah.

**Supervision:** Solomon A. Mensah.

**Visualization:** Zoe Vittum, Solomon A. Mensah.

**Writing – original draft:** Zoe Vittum, Solomon A. Mensah.

**Writing – review & editing:** Zoe Vittum, Solomon A. Mensah.

## Acknowledgments

We want to acknowledge the help of Udaya Rattan and Jacqueline O'Donnell for their help in data collection.

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
