## [Decision Letter · Decision Letter 0]

12 Sep 2025

Dear Dr. Vittum,

Thank you for submitting your manuscript to PLOS ONE. After careful consideration, we feel that it has merit but does not fully meet PLOS ONE’s publication criteria as it currently stands. Therefore, we invite you to submit a revised version of the manuscript that addresses all the points raised during the review process.

We look forward to receiving your revised manuscript.

Kind regards,

Mária A. Deli, M.D., Ph.D.

Academic Editor

PLOS ONE

Journal Requirements:

“This work was funded by startup funding awarded to Prof. Mensah by Worcester Polytechnic Institute. The authors have declared that no competing interests exist.”

Additional Editor Comments:

Two experts have evaluated the manuscript. They agreed it has merits but further amendments are needed. Please complete the asked methodological parts, and the discussion. Further lectin staining is suggested to get more data on glycan diversity behind the observed changes.

Reviewers' comments:

Reviewer's Responses to Questions

**Comments to the Author**

1. Is the manuscript technically sound, and do the data support the conclusions?

Reviewer #1: Yes

Reviewer #2: Yes

2. Has the statistical analysis been performed appropriately and rigorously?

Reviewer #1: Yes

Reviewer #2: Yes

3. Have the authors made all data underlying the findings in their manuscript fully available?

Reviewer #1: Yes

Reviewer #2: Yes

4. Is the manuscript presented in an intelligible fashion and written in standard English?

Reviewer #1: Yes

Reviewer #2: Yes

Reviewer #1: Summary: This manuscript presents a study of apical and basal glycocalyx expression in human lung microvascular endothelial cells (HLMVECs) under varying fluid shear stress conditions. The authors explore temporal changes in glycocalyx coverage, intensity, and thickness across a range of physiologically and pathophysiologically relevant fluid shear stress magnitudes. The work is timely and contributes meaningfully to our understanding of endothelial mechanobiology.

Strengths

1. The study includes multiple fluid shear stress magnitudes and time points (0 hr, 30 min, 12 hr), allowing for dynamic interpretation of glycocalyx behavior.

2. The focus on both apical and basal glycocalyx responses is valuable, especially the suggestion that basal glycocalyx may be sensitive to apical fluid shear stress.

3. The findings under very low fluid shear stress conditions may have important implications for vascular pathologies and warrant further exploration.

Weaknesses & Suggestions for Improvement

1. Cover Sheet Inconsistencies:

1.1. The funding disclosure states no support was received, yet institutional funding is mentioned in the manuscript. This should be corrected to ensure proper acknowledgment.

1.2. The data availability statement includes a minor typographical error: “analyzed the presented” likely should read “analyzed and presented.”

2. Introduction Clarity:

2.1. The mention of turbulence is vague. Does turbulent flow occur in the body, and under what conditions? Clarifying this would strengthen the physiological relevance.

2.2. The hypothesis is not explicitly stated. A clear hypothesis would help frame the study and guide interpretation of the results.

3. Methods Presentation:

3.1. Supplemental Figure 3 could be in the main body of the paper

4. Results Presentation:

4.1. The emphasis on permeabilization vs. non-permeabilization is confusing given that glycocalyx is extracellular. The rationale for this distinction should be clarified.

4.2. Figures 2 to 5 do not indicate whether cells were permeabilized.

4.3. In Figure 1, statistical comparisons across time points are missing. Given the observed decrease at 30 min and recovery at 12 hr, statistical analysis would strengthen the conclusions.

4.4. Supplemental figure data appear inconsistent with the purple bars in Figure 1, though both are said to represent non-permeabilized cells. This discrepancy should be explained.

5. Discussion Gaps:

5.1. There is a statement made that glycocalyx returns to baseline after 12 hr. The data shows 12 hr fluid shear stress leads to glycocalyx levels matching or falling below static conditions, depending on specific shear stress magnitude. This contradicts the expectation that fluid shear stress promotes glycocalyx growth and should be addressed.

5.2. The authors should elaborate on the implications of the finding that very low fluid shear stress drives a decrease in glycocalyx coverage and intensity over time, as this has important disease relevance.

5.3. The conclusion that basal glycocalyx is sensitive to apical fluid shear stress is intriguing but underexplored. The authors should expand on possible mechanisms and significance.

5.4. The observation that differences between fluid shear stress magnitudes only emerge at 12 hours (Figure 5) suggests short-term studies may be inadequate for HLMVECs. This point deserves more emphasis in the discussion.

6. Goal vs. Interpretation Misalignment:

6.1. The results are heavily focused on permeabilization status, which does not clearly align with the stated goal of examining glycocalyx expression. The authors should clarify why this distinction is central to their analysis.

6.2. Several findings appear to reconfirm prior published work. The authors should clarify what is novel in their observations and how their data extend or challenge existing literature.

Reviewer #2: The work submitted to PLOS One by Vittum & Mensah et al., entitled: “Fluid shear stress-dependent modulation of the basal endothelial glycocalyx.” Provides an innovative approach to measuring the glycocalyx composition on both sides of endothelial cells under various flow conditions. The authors provide an innovative method for measuring glycocalyx thickness in an endothelial cell culture system exposed to various flow conditions. The main issue with the manuscript is the lack of detail in the methods section. More specifically, the manuscript lacks enough references for the design and rationale. In its current form, the manuscript is not suitable for publication. The authors can respond to the comments below for future consideration. The issues mentioned above, as well as additional major and minor concerns, are outlined below:

Major:

1. The manuscript lacks sufficient details in the methodology, for example:

Line 348: Need source for 3-4 days being sufficient for full maturation of GCX on HLMVECs

Line 364 Please provide source which shows that WGA binds to ‘all lectins of the GCX’

Line 346 Can the authors elaborate on why media that is optimized for breast tissue is being used for lung endothelial cells?

Line 365 the use of streptavidin is confusing, as it is unclear where biotin is used in the methods

Line 389 The authors state that: “Background intensity was determined to be the intensity that corresponds to 6% of the normalized distribution of pixel intensities in the apical GCX and 8% in the basal GCX for each experimental group” Can the authors elaborate on how these 6 and 8 % values were determined.

2. The manuscript lacks depth as the same assessments were done across different conditions and turned into separate figures. The different FSS rates could be combined into a single figure. This would leave room for figures that focus on specific glycan changes under FSS conditions.

3. The manuscript could benefit from the addition of a lectin with specificity to a particular glycan component of the GCX (e.g., hyaluronan, or heparan sulfates). The use of a Pan-GCX marker does imply the change, but the mechanism of those changes could be parsed out with more specific markers for GCX components. This was mentioned on line 334, but at least some data should be presented showing specific changes in a particular glycan after FSS.

4. In the Introduction (line 43) and in the discussion, there are mentions of heparan sulfates and syndecan-1. These markers could have been assessed in this model system using validated probes for their expression.

5. Can the authors elaborate on how changes in cell polarity (i.e., movement of the nucleus, as demarcated by DAPI) could influence their interpretation of Apical vs. Basal surfaces under different flow conditions?

Minor:

Line 225: The use of the term peptides here seems vague. Please be more specific in the discussion.

Line 363 Should Wheat germ albumin be ‘agglutinin’?

Line 360 Bovine serum albumin (BSA) needs to be defined earlier

Figure legends should be placed at the end of the manuscript on initial submission, rather than embedded in the results section.

**Do you want your identity to be public for this peer review?** For information about this choice, including consent withdrawal, please see our Privacy Policy

Reviewer #1: No

Reviewer #2: No

---

## [Author Response · Author response to Decision Letter 1]

4 Nov 2025

We would like our funding statement to reflect the following statement: This work was funded by startup funding awarded to Prof. Solomon A. Mensah by Worcester Polytechnic Institute.

We believe all other concerns have been addressed in text or within the reviewer response letter. We greatly appreciate your feedback and how it has helped us refine our manuscript.

---

## [Decision Letter · Decision Letter 1]

19 Nov 2025

Dear Dr. Vittum,

Thank you for submitting your manuscript to PLOS ONE. After careful consideration, we feel that it has merit but does not fully meet PLOS ONE’s publication criteria as it currently stands. Therefore, we invite you to submit a revised version of the manuscript that addresses the points raised during the review process.

We look forward to receiving your revised manuscript.

Kind regards,

Mária A. Deli, M.D., Ph.D.

Academic Editor

PLOS ONE

Journal Requirements:

Reviewers' comments:

Reviewer's Responses to Questions

**Comments to the Author**

Reviewer #1: (No Response)

Reviewer #2: All comments have been addressed

2. Is the manuscript technically sound, and do the data support the conclusions?

Reviewer #1: Yes

Reviewer #2: Yes

3. Has the statistical analysis been performed appropriately and rigorously?

Reviewer #1: Yes

Reviewer #2: Yes

4. Have the authors made all data underlying the findings in their manuscript fully available?

Reviewer #1: Yes

Reviewer #2: Yes

5. Is the manuscript presented in an intelligible fashion and written in standard English?

Reviewer #1: Yes

Reviewer #2: Yes

Reviewer #1: The authors have been responsive to both sets of reviewer comments, and the revisions have substantially improved the manuscript. Most of the weaknesses noted in the initial review have been addressed, which is commendable.

Two minor concerns remain:

Figures 2–5: The figure legends do not clearly indicate whether the cells were permeabilized. If this information is already included, it should be made explicit in the legends or methods section for clarity.

Figure 1: Statistical comparisons across time points are still missing. Given the observed decrease at 30 minutes and recovery at 12 hours, statistical analysis would strengthen the conclusions. In particular, for panels H and I, statistical analysis across time points appears feasible unless the authors consider it unnecessary due to the detailed data presented in Figures 2–4. If so, a brief clarification would be helpful.

Overall, the manuscript is much improved, and addressing these final points will further enhance clarity and rigor.

Reviewer #2: The authors have adequately addressed reviewer comments. The manuscript is much improved and suitable for publication

**Do you want your identity to be public for this peer review?** For information about this choice, including consent withdrawal, please see our Privacy Policy

Reviewer #1: No

Reviewer #2: **Yes:** Aric F. Logsdon

---

## [Author Response · Author response to Decision Letter 2]

25 Nov 2025

We believe we have addressed the remaining two reviewer concerns in the reviewer response letter and that there is no outstanding request or concerns from the editor, especially with the addition of the data sets. Once again, we greatly appreciate your time and careful review of our manuscript.

---

## [Decision Letter · Decision Letter 2]

7 Dec 2025

Fluid shear stress-dependent modulation of the basal endothelial glycocalyx

PONE-D-25-37102R2

Dear Dr. Vittum,

We’re pleased to inform you that your manuscript has been judged scientifically suitable for publication and will be formally accepted for publication once it meets all outstanding technical requirements.

Kind regards,

Mária A. Deli, M.D., Ph.D.

Academic Editor

PLOS One

Additional Editor Comments (optional):

Reviewers' comments:

Reviewer's Responses to Questions

**Comments to the Author**

Reviewer #1: All comments have been addressed

2. Is the manuscript technically sound, and do the data support the conclusions?

Reviewer #1: Yes

3. Has the statistical analysis been performed appropriately and rigorously?

Reviewer #1: Yes

4. Have the authors made all data underlying the findings in their manuscript fully available?

Reviewer #1: Yes

5. Is the manuscript presented in an intelligible fashion and written in standard English?

Reviewer #1: Yes

Reviewer #1: Previous critique #1 has been satisfactorily addressed. Regarding critique #2 (“Figure 1: Statistical comparisons across time points are still missing”), this point was not directly addressed in the revision. However, this is a minor concern since the necessary comparisons across time points are demonstrated in other figures within the manuscript.

**Do you want your identity to be public for this peer review?** For information about this choice, including consent withdrawal, please see our Privacy Policy

Reviewer #1: No

---

## [Editor Report · Acceptance letter]

PONE-D-25-37102R2

PLOS One

Dear Dr. Vittum,

I'm pleased to inform you that your manuscript has been deemed suitable for publication in PLOS One. Congratulations! Your manuscript is now being handed over to our production team.

Kind regards,

on behalf of

Prof. Mária A. Deli

Academic Editor

PLOS One